# Multi-Year Comparison of $CO_2$ Concentration from NOAA Carbon Tracker Reanalysis Model with Data from GOSAT and OCO-2 over Asia

**Farhan Mustafa** [1], **Lingbing Bu** [1,*], **Qin Wang** [1], **Md. Arfan Ali** [2], **Muhammad Bilal** [2], **Muhammad Shahzaman** [3] and **Zhongfeng Qiu** [2]

[1]  Collaborative Innovation Center on Forecast and Evaluation of Meteorological Disasters, Key Laboratory for Aerosol-Cloud-Precipitation of China Meteorological Administration, Key Laboratory of Meteorological Disasters, Ministry of Education, Nanjing University of Information Science and Technology, Nanjing 210044, China; farhan@nuist.edu.cn (F.M.); 20181205013@nuist.edu.cn (Q.W.)

[2]  School of Marine Sciences, Nanjing University of Information Science and Technology, Nanjing 210044, China; md.arfanali@nuist.edu.cn (M.A.A.); muhammad.bilal@connect.polyu.hk (M.B.); zhongfeng.qiu@nuist.edu.cn (Z.Q.)

[3]  School of Atmospheric Sciences, Nanjing University of Information Science and Technology, Nanjing 210044, China; mshahzaman786@nuist.edu.cn

*  Correspondence: lingbingbu@nuist.edu.cn; Tel.: +86-189-519-97222

**Abstract:** Accurate knowledge of the carbon budget on global and regional scales is critically important to design mitigation strategies aimed at stabilizing the atmospheric carbon dioxide ($CO_2$) emissions. For a better understanding of $CO_2$ variation trends over Asia, in this study, the column-averaged $CO_2$ dry air mole fraction ($XCO_2$) derived from the National Oceanic and Atmospheric Administration (NOAA) CarbonTracker (CT) was compared with that of Greenhouse Gases Observing Satellite (GOSAT) from September 2009 to August 2019 and with Orbiting Carbon Observatory 2 (OCO-2) from September 2014 until August 2019. Moreover, monthly averaged time-series and seasonal climatology comparisons were also performed separately over the five regions of Asia; i.e., Central Asia, East Asia, South Asia, Southeast Asia, and Western Asia. The results show that $XCO_2$ from GOSAT is higher than the $XCO_2$ simulated by CT by an amount of 0.61 ppm, whereas, OCO-2 $XCO_2$ is lower than CT by 0.31 ppm on average, over Asia. The mean spatial correlations of 0.93 and 0.89 and average Root Mean Square Deviations (RMSDs) of 2.61 and 2.16 ppm were found between the CT and GOSAT, and CT and OCO-2, respectively, implying the existence of a good agreement between the CT and the other two satellites datasets. The spatial distribution of the datasets shows that the larger uncertainties exist over the southwest part of China. Over Asia, NOAA CT shows a good agreement with GOSAT and OCO-2 in terms of spatial distribution, monthly averaged time series, and seasonal climatology with small biases. These results suggest that $CO_2$ can be used from either of the datasets to understand its role in the carbon budget, climate change, and air quality at regional to global scales.

**Keywords:** CarbonTracker; GOSAT; OCO-2; $XCO_2$; Asia; greenhouse gases

## 1. Introduction

Dealing with global climate change requires accurate knowledge of the global budget of atmospheric greenhouse gases [1]. Atmospheric carbon dioxide ($CO_2$) is an important greenhouse gas that plays a significant role in several atmospheric phenomena such as hydrology, sea ice melting, sea level increasing and most importantly the temperature [2,3]. Since the industrial revolution, a notable increase in global $CO_2$ concentration has been observed from 280 ppm in the middle of the

19th century to over 400 ppm to date [4–6]. Combustion of fossil fuel, land-use changes, and cement production are the major causes of increased $CO_2$ concentration [7].

Asia, with the world's 10 largest $CO_2$ emitting nations, has become increasingly important in studying the global carbon budget. The poverty elimination efforts by the rapidly growing economies of the continent have largely relied on the construction of fossil-fuel based power and industrial projects which have significantly contributed to the greenhouse gases emissions [8]. Knowledge of regional contributions of $CO_2$ is critically important to design mitigation strategies aimed at stabilizing the $CO_2$ emissions. Much efforts have been made for estimating carbon sources and sinks in Asia, but still, they remain poorly quantified [9,10]. A detailed budget of $CO_2$ exchange between the earth's surface and the atmosphere is not available for some regions like South Asia, due to a sparse network of key carbon observations [11]. Moreover, Asia also remains one of the regions with large uncertainties in the carbon budget. In particular, the boreal and eastern part of Asia produces large uncertainties because of the large land surface heterogeneity and complex interactions between biosphere and atmosphere [12]. There is a need to expand the atmospheric observation network and develop inverse modelling systems for the effective use of in situ and remote sensing data streams [13].

A comprehensive network of ground-based, sun-viewing, near-IR, Fourier transform spectrometers known as Total Carbon Column Observing Network (TCCON) has been established to accurately measure the atmospheric greenhouse gases such as $CO_2$, $CO$, $N_2O$, and $CH_4$ [14,15]. A significant effort has been applied and the $CO_2$ measurements taken by TCCON have resulted in a precision of 0.25% under clear atmospheric conditions [16]. However, because of limited spatial coverage and uneven distribution of the TCCON sites, accurate $CO_2$ amounts cannot be measured on sub-continental and regional spatial scales [17]. Studies suggest that the satellite observations, with their lower precision than the ground-based measurements but higher spatial coverage, can help to improve the $CO_2$ measurements [18–20].

Over the past decade, multiple satellites such as the Greenhouse Gases Observing Satellite (GOSAT), Orbiting Carbon Observatory 2 (OCO-2) [21,22] and TanSat [23,24] have been dedicated to improving the $CO_2$ measurements and covering the spatiotemporal gaps. GOSAT and OCO-2 have a common observational approach. For example, both the satellites have the solar reflectance spectra centered around 1.6 and 2.0 μm which are used to determine the $CO_2$ optical depth and band A which is centered around 0.76 μm to measure the $O_2$ optical depth. The information from these three spectral regions is combined to determine the $XCO_2$ retrieval. $XCO_2$ is a column-averaged $CO_2$ dry air mole fraction that shows spatiotemporal variability and helps in improving the estimations of $CO_2$ concentrations and constraining the model simulations by minimizing the uncertainties of $CO_2$ sources and sinks. It has been shown that in the presence of optically thin clouds or aerosols, neglecting scattering can lead to unacceptably large retrieval errors [25]. Several retrieval algorithms, such as the National Institute of Environment Studies (NIES) $XCO_2$ retrieval algorithm [26], University of Leicester full physics retrieval algorithm [27], and NASA's Atmospheric $CO_2$ Observations from Space (ACOS) $XCO_2$ retrieval algorithm [28,29] have devised different approaches to deal with the scattering effects in the retrieval of the $CO_2$. GOSAT contains the Clouds and Aerosol Imager (CAI) to detect the clouds but OCO-2 does not have any dedicated instrument to detect the clouds and aerosols. Although the satellite data are corrected (retrievals with the large error are screened out), uncertainties remain in the data. Therefore, the satellite datasets are validated against precise datasets to ensure the data quality and compared with the other datasets to assess the consistency and potential for joint utilization.

The CarbonTracker (CT), originally developed by National Oceanic and Atmospheric Administration (NOAA), is a modern data assimilation system that integrates the data into a consistent estimate of surface $CO_2$ exchange [30]. CT incorporates daily $CO_2$ measurements derived from the network of tall towers calibrated according to the world $CO_2$ standard. Better knowledge and understanding of regional and global carbon cycles depend on the accuracy of the $CO_2$ estimations. Therefore, the satellite and the model data need to be validated against other satellite observations and/or in situ observations before using them to answer scientific questions.

Many inter-comparisons and validation studies have reported discrepancies in regional and global carbon budgets and their spatial distributions because of differences in their set of criteria, approaches, assumptions, and accuracy of the available data. Because of quality control purposes, much of the space-based observations are screened out during pre-processing. Oshchepkov et al. [31] pointed out that less than 10% of the total measurements become usable after retrieval and the screening processing. This might be due to the high amount of aerosol which significantly limits $XCO_2$ coverage. Deng et al. [32] compared the NIES and ACOS retrieval algorithms results for GOSAT with the TCCON site and reported the overall bias of $0.21 \pm 1.85$ ppm and $-0.69 \pm 2.13$ ppm for ACOS and NIES $XCO_2$, respectively. Moreover, the greatest monthly mean difference of $1.43 \pm 0.60$ ppm lies over China. Jing et al. [33] compared the $CO_2$ concentration simulated by GEOS-Chem with GOSAT, CT, and TCCON. GEOS-Chem is overestimated by an amount of 0.78 ppm when compared with GOSAT, slightly underestimated when compared with the CT, and underestimated by an amount of 1 ppm when compared with the TCCON. Kulawik et al. [34] compared the multiple $CO_2$ data sources with the TCCON and found mean root square deviations of 1.7 and 0.9 in GOSAT and CT2013x $XCO_2$ retrievals, respectively. These findings suggest that there is an imminent need to assess the accuracy and uncertainty of $XCO_2$ observations derived from the space-based data against in-situ measurements and accurate simulations by models. Moreover, the inter-comparisons between the models and other $CO_2$ data sources are also essential to quantitatively evaluate the model uncertainties and further improve the results.

This study aims to inter-compare the CT $CO_2$ characteristics with ACOS/GOSAT and OCO-2 $XCO_2$ retrievals over Asia and separately on its five regions i.e., Central Asia, East Asia, South Asia, Southeast Asia, and Western Asia from September 2009 to August 2019. Furthermore, the ability and consistency of the CarbonTracker to capture the seasonal and inter-annual variations of $CO_2$ amplitude over different regions of Asia have been assessed.

## 2. Materials and Methods

### 2.1. Study Area

Asia covers one-third of the Earth's surface and almost every known climate occurs on this continent [35]. The climatic heterogeneity in Asia is driven by several factors such as humidity, temperature, precipitation, spatial distribution and intensity of solar energy, wind pattern and atmospheric pressure. $CO_2$ fluxes observe seasonal variability and to determine the regional variability, Asia has been divided into five regions (Figure 1). The region names and the countries are given in the following:

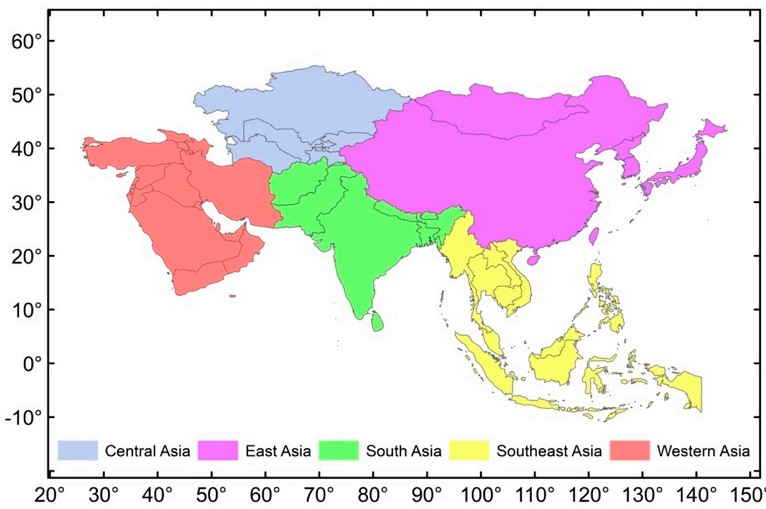

**Figure 1.** Regional division of Asia.

This study is carried out over Asia as a whole and separately in each of its regions, given in Table 1.

**Table 1.** The table shows the regional distribution of the study area.

| No. | Region | Countries |
|-----|--------|-----------|
| 1 | Central Asia | Uzbekistan, Turkmenistan, Tajikistan, Kyrgyzstan, Kazakhstan |
| 2 | East Asia | China, Japan, Hong Kong, Macau, North Korea, South Korea, Taiwan, Mongolia |
| 3 | South Asia | Pakistan, Bangladesh, Afghanistan, India, Sri Lanka, Nepal, Bhutan, Maldives |
| 4 | Southeast Asia | Singapore, Thailand, Malaysia, Indonesia, Philippine, Brunei, Myanmar, Cambodia, Laos, Vietnam, Timor-Leste |
| 5 | Western Asia | Armenia, Azerbaijan, Bahrain, Cyprus, Georgia, Iran, Iraq, Israel, Jordan, Kuwait, Lebanon, Oman, Palestine, Qatar, Saudi Arabia, Syria, Turkey, United Arab Emirates, Yemen |

*2.2. Datasets*

2.2.1. GOSAT Observations

GOSAT, the world's first satellite dedicated to measuring the concentrations of $CO_2$ and $CH_4$ from space, is a joint venture of Japan Aerospace Exploration Agency (JAXA), NIES, and Ministry of Environment (MOE). GOSAT was launched on 23 January 2009 and it takes measurements of reflected sunlight in three near-infrared bands with a circular footprint of approximately 10.5 km diameter at the nadir [29,36,37]. It incorporates two instruments: the Thermal and Near-infrared Sensor for carbon Observation-Fourier Transform Spectrometer (TANSO-FTS), and the Thermal and Near-infrared Sensor for carbon Observation-Cloud and Aerosol Imager (TANSO-CAI). TANSO-FTS is responsible for measuring the greenhouse gases with three narrow SWIR bands (0.76, 1.6 and 2.0 μm) and a wide thermal band (5.5–14.3 μm) at a spectral resolution of 0.2–1 cm. The SWIR bands are mainly responsible for retrieving $CO_2$ column concentrations while the TIR band retrieves the vertical profiles of $CO_2$ concentrations [38]. Clouds and aerosols strongly affect the quality of $CO_2$ observations [39]. TANSO-CAI retrieves the cloud and aerosol information [40,41]. In this study, the ACOS/GOSAT $XCO_2$ version 9.3 Level 2 Lite product has been used [16,28,29]. ACOS/GOSAT $XCO_2$ has lower bias and better consistency compared to NIES/GOSAT [32].

2.2.2. OCO-2 Observations

NASA's Orbiting Carbon Observatory-2 is the second satellite after GOSAT dedicated to monitoring atmospheric $CO_2$ to obtain better knowledge and understanding of the carbon cycle. The main objectives of the mission include measuring $CO_2$ with enough precision, accuracy, and spatial and temporal resolutions required to quantify the $CO_2$ sources and sinks at regional and global levels. The sun-synchronous, near-polar satellite incorporates three high-resolution spectrometers making coincident measurements of reflected sunlight in the near-infrared $CO_2$ near 1.61 and 2.06 μm and molecular oxygen ($O_2$) A-Band at 0.76 μm with a temporal resolution of 16 days allowing the complete global coverage of $XCO_2$ twice per month [27,42–44]. In this study, OCO-2 $XCO_2$ version 9r Level 2 Lite product has been used. These data were produced by OCO-2 project at the Jet Propulsion Laboratory, California Institute of Technology, and obtained from OCO-2 data archive maintained at the NASA Goddard Earth Science Data and Information Services Center.

2.2.3. CarbonTracker Measurements

CarbonTracker (CT), developed by Peters et al. [45,46], is an inverse modelling framework that updates the atmospheric $CO_2$ distribution and the surface fluxes annually. It incorporates the two-way nested Transport Model 5 (TM5) offline atmospheric tracer transport model which supports high-resolution data regionally and coarse resolution data globally [47]. CT forecasts the $CO_2$ mole fraction by combing $CO_2$ surface exchange models and TM5 model driven by the meteorology from the European Center for Medium-Range Weather Forecasts (ECMRWF) ERA-interim reanalysis [48]. The resulting

three-dimensional $CO_2$ distribution is then sampled at the time and location that observations are available, and the difference between observations and model forecasts is minimized [30]. The CT provides the global $CO_2$ distribution at 25 pressure levels with the spatial resolution of $3° \times 2°$ Longitude/Latitude and temporal resolution of 3 h.

In this study, we have used CT2019 release and near real-time version (CT19NRT20). Previous versions of the modelled data have been compared with other $CO_2$ data sources and results suggest that CT captures the $XCO_2$ reasonably well. More detail about CT data can be accessed at (https://www.esrl.noaa.gov/gmd/ccgg/CT/).

*2.3. Methods*

Different $CO_2$ data products cannot be compared directly because of their differences in sampling methods and data sources. For instance, $CO_2$ products retrieved from OCO-2 and GOSAT are $CO_2$ column-averaged dry air mole fraction ($XCO_2$) concentrations while the simulated results from the CarbonTracker are $CO_2$ profiles with 25 vertical profiles. Moreover, the CT model gives well distributed smooth data with a temporal resolution of 3 h and a spatial resolution of $3° \times 2°$ Longitude/Latitude, while the satellite observations are different in terms of time and space. In satellite data, the screening process and cloud filtering significantly reduce the data quantity, which consequently produces gaps. In addition, the temporal resolution also contributes to mismatching between the model and the satellite data. As a result, both of the datasets cannot be compared directly because there is no match between the two datasets in terms of spatial and temporal resolutions. Thus, the CT mole fraction of $CO_2$ is extracted on the time and location of the satellite data. By keeping the grid points of CT as standard, the corresponding $XCO_2$ from the satellite observations are extracted for a spatial window of $1.5° \times 1.5°$ with their centers at the reference grid points of the CT. Furthermore, to make both the datasets comparable, it is mandatory to adjust the vertical resolutions of the datasets accordingly. To deal with the different number of vertical levels of both datasets, CT $CO_2$ is interpolated to the vertical numbers of the satellite datasets. The CT $XCO_2$ ($XCO_2^m$), that is used to compare with the satellite datasets is computed by the procedure as suggested by [33,49,50].

$$XCO_2^m = XCO_2^a + \sum_j p_j^T K_j * (CO_2^i - CO_{2a}), \tag{1}$$

where, ($XCO_2^a$) is $XCO_2$ a priori, (p) is pressure weighting function, (K) is an averaging kernel, ($CO_2^i$) is interpolating CT $CO_2$, ($CO_{2a}$) is a priori profile, (j) refers to satellite retrieval vertical level, and (T) indicates the matrix transpose. Some other statistical calculations such as correlation coefficients (R), and Root Mean Square Deviation (RMSD) have also been computed to determine the level of agreement between the CT and respective satellite datasets. The correlation coefficient for the *j*th pixel is computed by the formula given:

$$R_j = \frac{\frac{1}{n}\sum_{i=1}^{n}(M_i - \overline{M})(\frac{1}{n}\sum_{i=1}^{n}(S_i - \overline{S}),}{\sqrt{\frac{1}{n}\sum_{i=1}^{n}(M_i - \overline{M})^2}\sqrt{\frac{1}{n}\sum_{i=1}^{n}(S_i - \overline{S})^2}} \tag{2}$$

where ($\overline{M}$) and ($\overline{S}$) represent the mean values of CT and satellite $XCO_2$ retrievals. The RMSD showing the standard error of the model against the observations at the *j*th pixel is calculated as follows:

$$RMSD_j = \sqrt{\frac{1}{n}\sum_{i=1}^{n}((M_i - \overline{M}) - (S_i - \overline{S}))^2} \tag{3}$$

More detail about the Root Mean Squared Deviation (RMSD) is given in [51].

## 3. Results and Discussion

### 3.1. Comparison of NOAA CarbonTracker with GOSAT and OCO-2

NOAA CarbonTracker has been compared with ACOS/GOSAT version 9.3 and OCO-2 version 9r for durations of 10 years starting from September 2009 to August 2019 and 5 years ranging from September 2014 to August 2019, respectively. The CT dataset is made comparable with the satellite datasets following the procedure described in Section 2.2 The results are based on 519 points which are uniformly distributed over Asia with a spatial resolution of $3° \times 2°$ Longitude/Latitude.

Figure 2 shows the distribution of 10 years averaged $XCO_2$ over Asia for CT (Figure 2a) and GOSAT (Figure 2b). The number of datasets used in the analysis is shown in Figure 2d. The datasets in each grid range from 11 to 2239. The comparison shows that the spatial distribution characteristics are consistent between both of the datasets with a spatial mean correlation of 0.93.

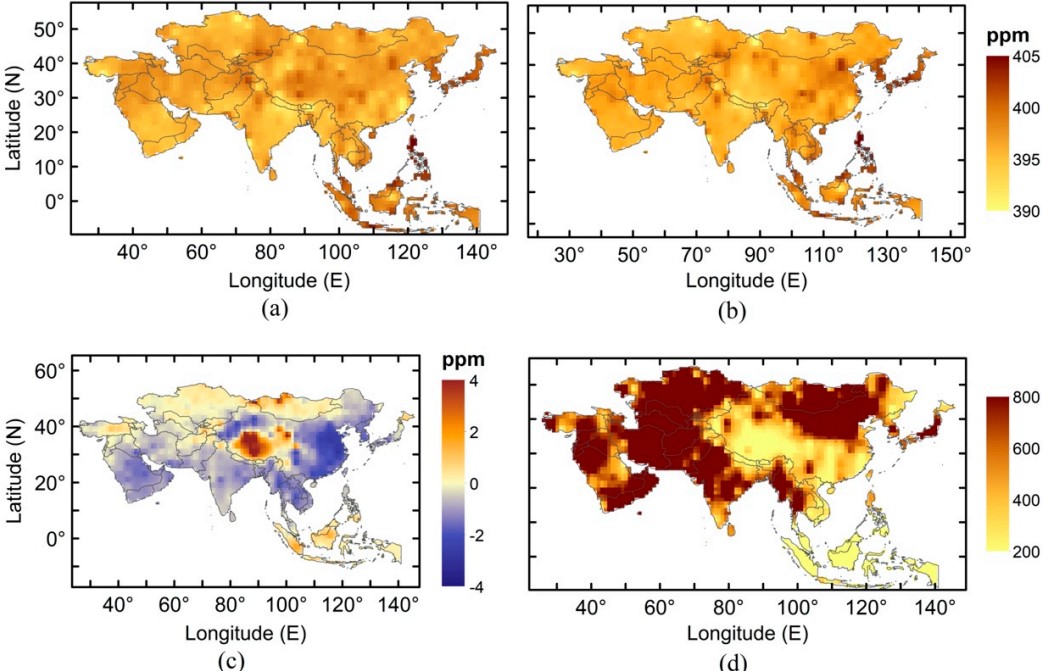

**Figure 2.** (**a**) Distribution of 10 years mean $XCO_2$ over Asia for (**a**) CarbonTracker (CT) 2019; (**b**) Greenhouse Gases Observing Satellite (GOSAT); (**c**) their differences (CT-GOSAT); and (**d**) the total number of datasets from GOSAT in each grid.

Figure 3 shows the distribution of temporally averaged $XCO_2$ derived from the CT (Figure 3a) and OCO-2 (Figure 3b) over Asia for 5 years ranging from September 2014 to August 2019. Southeast Asia and some parts of China observe the least number of cloud-free observations from OCO-2 (Figure 3d). The two datasets show good consistency with a spatial mean correlation of 0.89. OCO-2 has better spatial coverage than GOSAT over Asia (Figures 2d and 3d), however, both show a small number of retrievals over East Asia and Southeast Asia. This may be due to the ACOS $XCO_2$ retrieval algorithm which excludes the data with high aerosol optical depth and cloud optical thickness [28,29] and results in reducing $XCO_2$ data quantity over Asian regions during the monsoon [52] and high pollution events, which typically occur in the spring [53]. Both the model and the satellite datasets observe the highest $CO_2$ emission over high density urban regions such as Beijing-Tianjin-Hebei area in northern China, Korea and Japan (Figure 2a,b and Figure 3a,b). These are the most populated regions with the largest anthropogenic emissions in the world [54,55]. The distribution characteristics of $XCO_2$ values retrieved from GOSAT and OCO-2 are similar to the modelled $XCO_2$ over most of the regions except some parts of East Asia such as China and Mongolia (Figures 2c and 3c). GOSAT dataset shows a

higher standard deviation compared to OCO-2, and these results are supported by a previous study which has also reported a large standard deviation of GOSAT observations compared to OCO-2 at the TCCON sites [56]. The differences over China maybe mostly attributed to the large uncertainties in the terrestrial flux estimation as previous studies have reported poor performance of terrestrial flux from CT2015 against in situ observations obtained from eight sites of the Chinese Terrestrial Ecosystem Flux Observation and Research Network (ChinaFLUX) [57–60]. The substantial difference exists over the Qinghai Tibet Plateau, which is the highest terrain of the earth located in the southwest of China. The CT shows higher $XCO_2$ concentrations over mountain ranges of northern Mongolia, northern Afghanistan, Tajikistan and Kyrgyzstan compared to OCO-2 (Figure 3c). Kong et al. [56] also found significantly larger values of the CarbonTracker $XCO_2$ over Tibetan Plateau and mountain ranges compared to those of the satellite datasets.

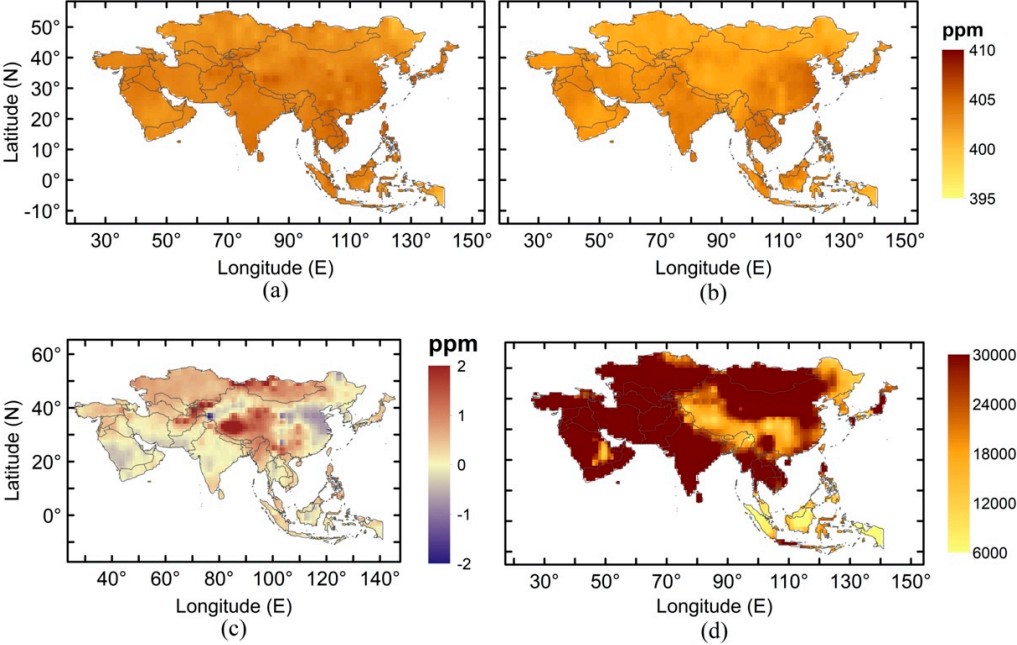

**Figure 3.** Distribution of 5 years mean $XCO_2$ over Asia (**a**) from CT; (**b**) Orbiting Carbon Observatory 2 (OCO-2); (**c**) their differences (CT-OCO2); (**d**) and the total number of datasets from OCO-2 in each grid.

Compared to the satellite datasets, CT is overestimated by −0.61 ± 0.96 ppm against GOSAT and underestimated by 0.31 ± 0.71 ppm against OCO-2 over Asia. Furthermore, the summaries of both comparisons including the regional detail are given in Tables 2 and 3.

**Table 2.** The table shows the statistical relationship between CT and GOSAT. The statistical parameters shown in the table are the mean correlation coefficient (R), an average of the root mean square deviation, the difference between the CT and GOSAT (D) with the standard deviation, the standard deviation in CT, the standard deviation in GOSAT and GOSAT error, which is GOSAT posterior estimate of $XCO_2$ error. The total number of datasets used in the analysis are 421,361 over 519 points distributed uniformly over Asia.

| Region | R | RMSD | D (Std.) | Std. in CT | Std. in GOSAT | GOSAT Error |
|---|---|---|---|---|---|---|
| Asia | 0.93 | 2.61 | −0.61 ± 0.96 | 2.59 | 2.62 | 1.19 |
| Central Asia | 0.96 | 1.87 | 0.03 ± 0.47 | 2.13 | 2.17 | 1.17 |
| East Asia | 0.87 | 3.96 | 0.32 ± 1.60 | 2.45 | 2.73 | 1.32 |
| South Asia | 0.94 | 2.34 | −0.67 ± 0.64 | 2.94 | 2.99 | 1.16 |
| Southeast Asia | 0.95 | 2.19 | −0.43 ± 1.01 | 3.65 | 3.49 | 1.02 |
| Western Asia | 0.96 | 1.87 | 0.02 ± 0.48 | 2.12 | 2.17 | 1.17 |

**Table 3.** The table shows the statistical relationship between the CT and OCO-2. The statistical parameters shown in the table are, the mean correlation coefficient (R), an average of the root mean square derivation, the difference between the CT and OCO-2 (D) with the standard deviation, the standard deviation in CT, the standard deviation in OCO-2 and OCO-2 error. The total number of datasets used in the analysis are 20,574,385 over 519 points distributed uniformly over Asia.

| Region | R | RMSD | D (Std.) | Std. in CT | Std. in OCO-2 | OCO-2 Error |
|---|---|---|---|---|---|---|
| Asia | 0.89 | 2.16 | 0.31 ± 0.71 | 1.07 | 1.10 | 0.62 |
| Central Asia | 0.93 | 2.04 | 0.72 ± 0.64 | 0.92 | 0.65 | 0.60 |
| East Asia | 0.84 | 2.94 | 0.33 ± 0.92 | 1.30 | 1.31 | 0.70 |
| South Asia | 0.89 | 2.09 | 0.25 ± 0.73 | 0.72 | 0.67 | 0.63 |
| Southeast Asia | 0.93 | 1.54 | 0.29 ± 0.39 | 1.78 | 1.19 | 0.53 |
| Western Asia | 0.94 | 1.49 | 0.06 ± 0.34 | 0.55 | 0.49 | 0.57 |

As a one-to-one comparison between CT model data and the satellite observations is not possible, it is therefore important to determine the relative spatial distance between the occurrences of both datasets. The satellite values are averaged in a 1.5° × 1.5° window centering the grid cell of CT. The spatial distance means a distance between a given GOSAT/OCO-2 measurement and CT grid point.

Figure 4 shows the scatter density plot between the CT and GOSAT (Figure 4a) and CT and OCO-2 (Figure 4b) $XCO_2$ concentrations. The color represents the relative spatial distance between the points of CT and the satellite datasets in terms of degree. The spatial distance between the points represents the spatial mismatch and does not indicate differences between the $XCO_2$ values of the datasets.

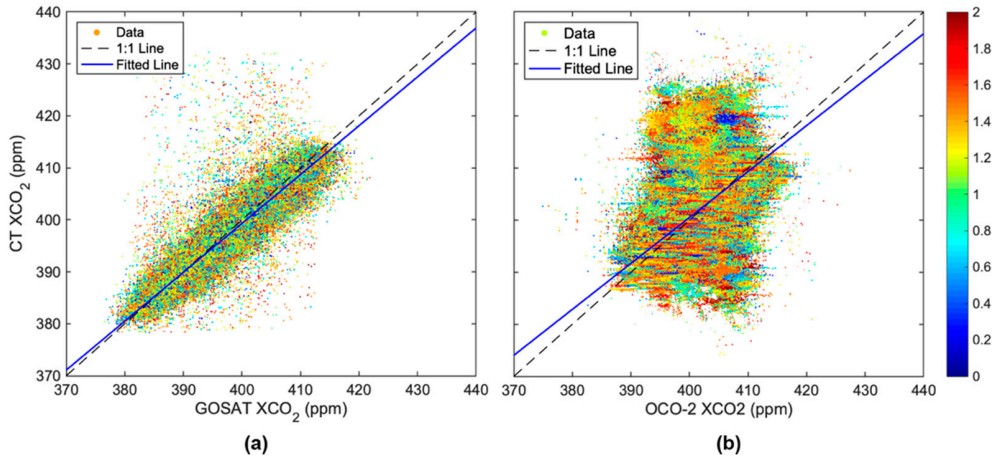

**Figure 4.** The scatter density plot between the $XCO_2$ derived from (**a**) the CT and GOSAT; (**b**) and the CT and OCO-2.

Figure 5 shows the spatial distribution of the correlations between the model and the satellite datasets. The results show a good correlation between CT and GOSAT (Figure 5a), and between CT and OCO-2 (Figure 5b) over Asia except for East Asia. However, a low correlation (<0.4) between CT and GOSAT is observed only over southwest China, while a low correlation (<0.4) between CT and OCO-2 is observed over southwest China, the northern part of Pakistan, northern Afghanistan, Tajikistan and Kyrgyzstan. These regions are either the plateaus or the mountain ranges. Both the model and the satellite datasets contribute to the uncertainties. Connor et al. [50] presented the OCO inverse method and prospective error analysis and concluded that the instrument can produce the single sounding error up to 0.8 ppm for sun conditions and up to 2.5 ppm for low sun conditions over the low- and mid-latitudes due to noise, geophysical and spectroscopic errors. To determine if it is the model data which originates the error or whether satellite data also contributes to it, the spatial distribution of the posterior $XCO_2$ error for GOSAT and OCO-2 is also shown in Figure 5c,d, respectively. The posterior

$XCO_2$ error is computed by combing interference errors, smoothing errors and instrument noise [50]. GOSAT posterior $XCO_2$ error is larger than that of OCO-2 and it becomes significant from East China to West China.

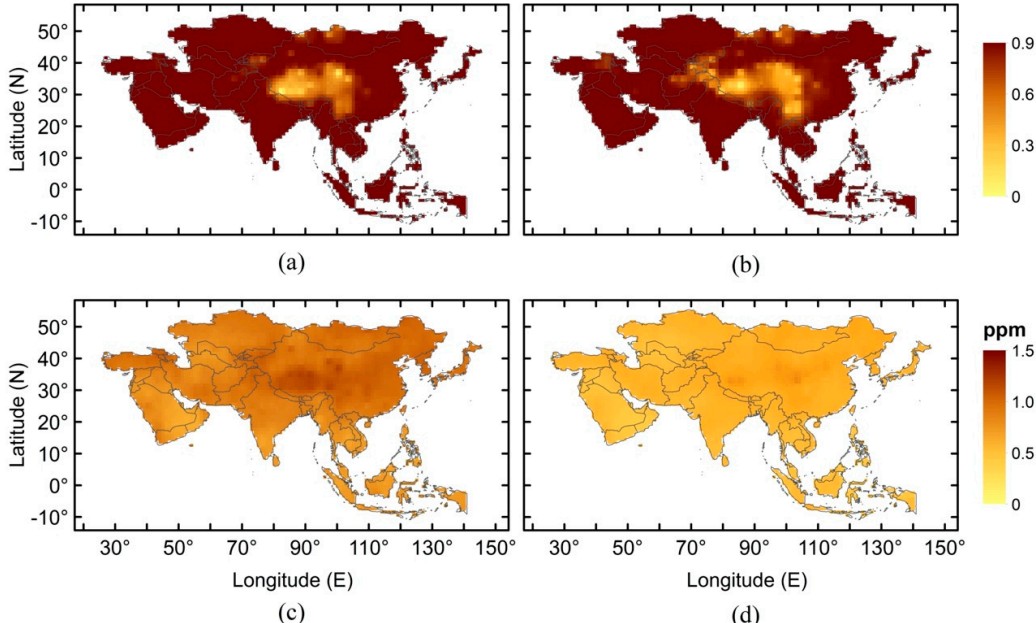

**Figure 5.** Spatial distributions of correlations between (**a**) the CarbonTracker and GOSAT; (**b**) the CarbonTracker and OCO-2; and mean posteriori estimate of $XCO_2$ uncertainty in (**c**) GOSAT; and (**d**) OCO-2.

In general, the CT-GOSAT comparison implies that both of the datasets are characterized by a high spatial mean correlation of 0.93, a global offset of −0.61 ppm, which suggests that CT $XCO_2$ is lower relative to that of GOSAT, RMSD of 2.61 ppm and relative accuracy of 0.96 ppm over Asia. The comparison between the CT and OCO-2 shows CT is overestimated by an amount of 0.31 ppm and exhibits a spatial mean correlation of 0.89, RMSD of 2.16 ppm and relative accuracy of 0.71 ppm. Furthermore, the summaries of both comparisons including the regional detail are given in Tables 2 and 3.

3.1.1. Monthly Averaged Time-Series Comparison

Figure 6 shows the monthly averaged $XCO_2$ variations from CT and the two satellite datasets for 5 years starting from September 2014 to August 2019 over Asia (Figure 6a), Central Asia (Figure 6b), East Asia (Figure 6c), South Asia (Figure 6d), Southeast Asia (Figure 6e), and Western Asia (Figure 6f). All the data products show similarity in phases and amplitudes with some differences in the magnitude, which suggests a good agreement between the model and the satellite datasets. The $CO_2$ concentrations over Central Asia (Figure 6b), East Asia (Figure 6c), and Southeast Asia (Figure 6e) are the highest in April and the lowest in August. It starts increasing in the fall, continues to increase in the winter, and reaches the maximum value in spring. In these regions, the heating systems are used in winter and spring which consume larger amounts of fossil energy, such as natural gas, oil, and coal, and thus produce a large amount of the $CO_2$ which is discharged into the atmosphere. Moreover, in winter and spring, the plants are in the dormant and recovery stage. During this period, the strong respiration and the weak photosynthesis also contribute significantly to increasing the $CO_2$ concentration in the atmosphere. The lower temperature in the winter inhibits the microorganism activity which hinders the decomposition process. The temperature starts increasing in the late spring, and it enhances the microorganism activity, thus the decomposition is started and $CO_2$ is released from the biological materials. This might be the reason for the maximum concentration of the $CO_2$ in April. The $CO_2$

concentration decreases from May to August. During this period, the temperature, precipitation, and the lightning are beneficial to the vegetation growth which enhances the photosynthesis process. The strong presence of the photosynthesis by vegetation decays $CO_2$ concentration [61,62]. Compared to the CT, GOSAT and OCO-2 show lower $CO_2$ concentrations during the wet season (April-August) over these regions.

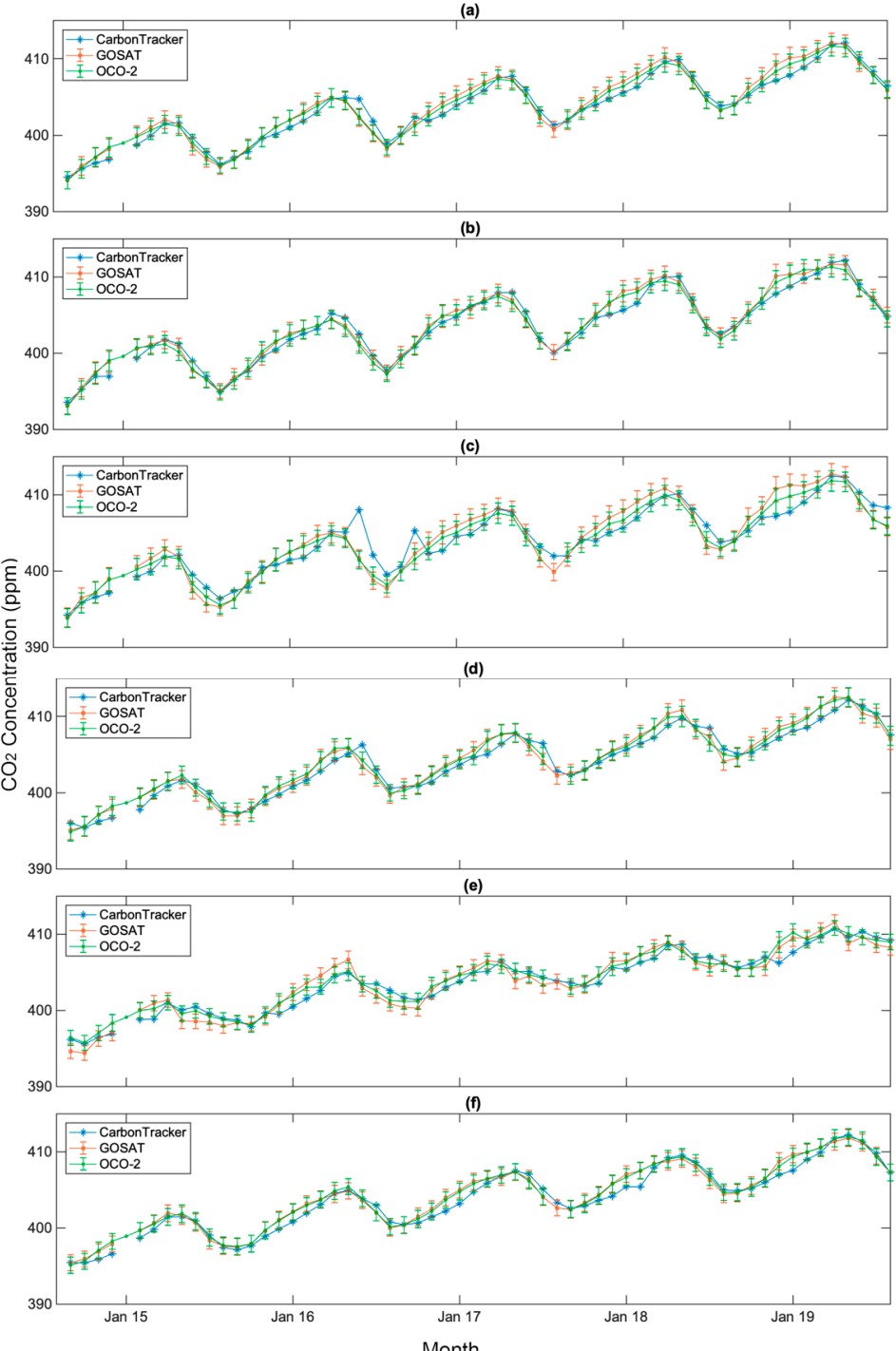

**Figure 6.** The time-series variations of monthly averaged $XCO_2$ derived from CT and the two satellite datasets for a period of 5 years ranging from September 2014 to August 2019 over (**a**) Asia; (**b**) Central Asia; (**c**) East Asia; (**d**) South Asia; (**e**) Southeast Asia; (**f**) and Western Asia, The gaps in the graph show the missing data.

Asia experiences a monsoon climate, which plays a significant role in the $CO_2$ concentrations [63]. Asian monsoon may be further classified into the Southwest Monsoon and the East Asian Monsoon [64]. The Southwest Monsoon mostly affects the Indian subcontinent and the East Asian Monsoon affects China, Japan, Philippines, Korea and Taiwan. The occurrence of the summer monsoons is a major source of precipitation and thus an important factor controlling the vegetation in these regions. Generally, March to May is treated as pre-monsoon, June to September as monsoon and October to November as post-monsoon in most of the Asian regions [63]. The $CO_2$ concentration is found to be maximum in the pre-monsoon and minimum in monsoon (Figure 6c,d).

The highest $CO_2$ concentration in the pre-monsoon maybe due to the higher temperature and solar radiation prevailing during these months, which stimulate the assimilation of $CO_2$ in the daytime and respiration in the night [65]. The low wind speed during the pre-monsoon causes slight mixing in the boundary layer which also contributes in the $CO_2$ concentration [66]. Moreover, a large amount of biomass burning and forest fires during this period also compliment the increase in $CO_2$ concentration [67,68]. The reduction in $CO_2$ concentration in monsoon is maybe because of the monsoon precipitation resulting in increasing the soil moisture which enhances the photosynthesis process. Moreover, the presence of clouds during the season decreases the temperature which reduces the leaf and soil respiration rate which eventually increases the carbon uptake [68]. The $CO_2$ starts increasing in post-monsoon which is associated with microbial activity and high ecosystem productivity [66,69]. The fluctuations in the CT $XCO_2$ have been observed during this season over East Asia and South Asia. Moreover, the CT shows higher $CO_2$ concentration compared to GOSAT and OCO-2 during monsoon (Tables 4 and 5). The higher $CO_2$ concentration over monsoon-affected regions might be due to the uncertainties involved in realizing the inversion during the monsoon season, in the presence of clouds [70].

**Table 4.** Monthly averaged $XCO_2$ concentration derived from the CT (C), GOSAT (G), and their differences (C-G). The statistics are computed for 5 years over Central Asia, East Asia, South Asia, Southeast Asia and Western Asia. (Unit: ppm).

| Month. | Central Asia | | | East Asia | | | South Asia | | | Southeast Asia | | | Western Asia | | |
|---|---|---|---|---|---|---|---|---|---|---|---|---|---|---|---|
| | C | G | C-G | C | G | C-G | C | G | C-G | C | G | C-G | C | G | C-G |
| Jan | 403.42 | 404.92 | −1.50 | 403.02 | 405.00 | −1.98 | 402.77 | 403.53 | −0.76 | 402.61 | 403.89 | −1.27 | 402.4 | 404.1 | −1.7 |
| Feb | 403.41 | 404.45 | −1.03 | 403.04 | 404.92 | −1.88 | 402.53 | 403.61 | −1.07 | 402.63 | 403.89 | −1.25 | 402.6 | 404.0 | −1.4 |
| Mar | 404.84 | 405.32 | −0.48 | 404.47 | 405.81 | −1.33 | 403.69 | 404.94 | −1.24 | 403.83 | 404.96 | −1.52 | 404.0 | 404.6 | −0.6 |
| Apr | 406.16 | 405.95 | 0.21 | 406.22 | 406.70 | −0.47 | 404.96 | 406.20 | −1.23 | 404.95 | 405.54 | −0.58 | 404.4 | 405.4 | −0.9 |
| May | 406.00 | 405.28 | 0.71 | 406.29 | 406.06 | 0.22 | 406.05 | 406.55 | −0.49 | 404.61 | 404.34 | 0.27 | 405.9 | 405.7 | 0.2 |
| Jun | 403.41 | 402.65 | 0.76 | 404.75 | 402.79 | 1.95 | 405.67 | 404.59 | 1.07 | 404.06 | 403.16 | 0.90 | 405.1 | 404.8 | 0.3 |
| Jul | 400.56 | 400.35 | 0.21 | 402.58 | 399.96 | 2.62 | 404.27 | 403.20 | 1.06 | 403.57 | 402.36 | 1.21 | 403.5 | 402.8 | 0.7 |
| Aug | 398.84 | 398.79 | 0.05 | 400.46 | 398.92 | 1.53 | 401.71 | 400.76 | 0.94 | 402.94 | 402.12 | 0.81 | 401.5 | 401.2 | 0.3 |
| Sep | 398.86 | 398.91 | −0.04 | 399.63 | 399.21 | 0.42 | 400.28 | 399.98 | 0.29 | 401.13 | 400.31 | 0.82 | 400.0 | 400.0 | 0 |
| Oct | 400.27 | 400.59 | −0.31 | 401.67 | 401.73 | −0.05 | 400.47 | 400.71 | −0.23 | 400.80 | 400.35 | 0.44 | 400.3 | 400.8 | −0.5 |
| Nov | 402.04 | 402.61 | −0.56 | 402.04 | 402.92 | −0.88 | 401.33 | 402.14 | −0.80 | 401.67 | 401.67 | 0.00 | 401.1 | 401.9 | −0.8 |
| Dec | 402.85 | 404.27 | −1.42 | 402.55 | 404.64 | −2.09 | 402.28 | 403.25 | −0.96 | 402.22 | 403.29 | −1.07 | 401.9 | 403.4 | −1.5 |

**Table 5.** Monthly averaged $XCO_2$ concentration derived from the CT (C), OCO-2 (O) and their differences (C-O). The statistics are computed for 5 years over Central Asia, East Asia, South Asia, Southeast Asia, and Western Asia. (Unit: ppm).

| Month | Central Asia | | | East Asia | | | South Asia | | | Southeast Asia | | | Western Asia | | |
|---|---|---|---|---|---|---|---|---|---|---|---|---|---|---|---|
| | C | O | C-O | C | O | C-O | C | O | C-O | C | O | C-O | C | O | C-O |
| Jan | 404.98 | 404.88 | 0.10 | 404.45 | 404.67 | −0.22 | 404.10 | 403.89 | 0.21 | 404.67 | 404.41 | 0.26 | 403.98 | 404.33 | −0.3 |
| Feb | 405.63 | 405.78 | −0.14 | 404.78 | 405.54 | −0.75 | 404.91 | 404.74 | 0.17 | 405.26 | 404.89 | 0.36 | 404.91 | 405.18 | −0.2 |
| Mar | 406.64 | 406.29 | 0.35 | 406.19 | 406.34 | −0.15 | 405.76 | 406.22 | −0.45 | 405.73 | 405.39 | 0.34 | 405.90 | 405.90 | 0 |
| Apr | 407.53 | 406.71 | 0.81 | 407.18 | 407.14 | 0.03 | 406.59 | 407.40 | −0.80 | 406.20 | 406.20 | 0 | 406.80 | 406.80 | 0 |
| May | 407.10 | 405.97 | 1.12 | 407.82 | 406.79 | 1.03 | 407.28 | 407.72 | −0.44 | 405.55 | 405.51 | 0.04 | 407.12 | 407.16 | −.04 |
| Jun | 404.71 | 403.64 | 1.07 | 405.85 | 404.08 | 1.77 | 406.70 | 406.19 | 0.50 | 405.26 | 404.79 | 0.46 | 406.43 | 406.18 | 0.25 |
| Jul | 402.07 | 401.40 | 0.66 | 405.55 | 401.81 | 3.73 | 406.24 | 404.64 | 1.60 | 404.75 | 404.24 | 0.50 | 404.81 | 404.16 | 0.65 |
| Aug | 400.20 | 399.58 | 0.62 | 403.06 | 400.65 | 2.41 | 404.67 | 402.49 | 2.17 | 404.27 | 403.76 | 0.50 | 402.91 | 402.44 | 0.46 |
| Sep | 398.95 | 398.60 | 0.34 | 400.58 | 399.25 | 1.33 | 400.90 | 399.90 | 1.00 | 401.46 | 400.90 | 0.55 | 400.30 | 400.02 | 0.27 |
| Oct | 400.62 | 400.51 | 0.11 | 401.46 | 401.07 | 0.39 | 400.84 | 400.50 | 0.34 | 401.47 | 400.79 | 0.67 | 400.61 | 400.61 | 0 |
| Nov | 402.59 | 402.54 | 0.05 | 401.76 | 402.44 | −0.68 | 401.93 | 402.03 | −0.09 | 402.52 | 402.10 | 0.41 | 401.71 | 401.92 | −0.2 |
| Dec | 403.82 | 404.28 | −0.45 | 402.96 | 403.99 | −1.02 | 403.15 | 403.21 | −0.06 | 403.49 | 403.56 | −0.06 | 403.10 | 403.32 | −0.2 |

The monthly averaged statistical relationships between the CT and satellite datasets are given in Tables 6 and 7. The CT is underestimated over all the regions of Asia when compared with GOSAT and overestimated over all the regions except East Asia when compared with OCO-2. East Asia exhibits lower correlation and higher RMSD values relative to other regions. The CT shows smaller RMSD with OCO-2 compared to GOSAT.

**Table 6.** The table shows the monthly averaged statistical relationship between the CT and GOSAT over Asia and its regions. The monthly averaged statistics are computed for 5 years starting from September 2014 to August 2019.

| Region | R | CT-GOSAT | RMSD | Dataset Quantity |
|---|---|---|---|---|
| Asia | 0.98 | −0.29 | 0.98 | 230,763 |
| Central Asia | 0.99 | −0.25 | 0.83 | 57,838 |
| East Asia | 0.94 | −0.11 | 1.74 | 74,982 |
| South Asia | 0.98 | −0.27 | 0.99 | 36,256 |
| Southeast Asia | 0.97 | −0.10 | 1.10 | 9974 |
| Western Asia | 0.98 | −0.37 | 0.86 | 51,713 |

**Table 7.** The table shows the statistical relationship between the CT and OCO-2 over Asia and its regions. The monthly averaged statistics are computed for 5 years starting from September 2014 to August 2019.

| Region | R | CT-OCO2 | RMSD | Dataset Quantity |
|---|---|---|---|---|
| Asia | 0.98 | 0.37 | 0.49 | 25,841,330 |
| Central Asia | 0.99 | 0.38 | 0.43 | 5,677,871 |
| East Asia | 0.94 | −0.63 | 1.09 | 6,869,504 |
| South Asia | 0.98 | 0.31 | 0.66 | 5,171,216 |
| Southeast Asia | 0.99 | 0.34 | 0.36 | 1,306,978 |
| Western Asia | 0.99 | 0.03 | 0.27 | 6,815,761 |

Figure 7 shows the annual growth of $XCO_2$ derived from the model and the two satellite datasets over Asia, Central Asia, East Asia, South Asia, Southeast Asia, and Western Asia.

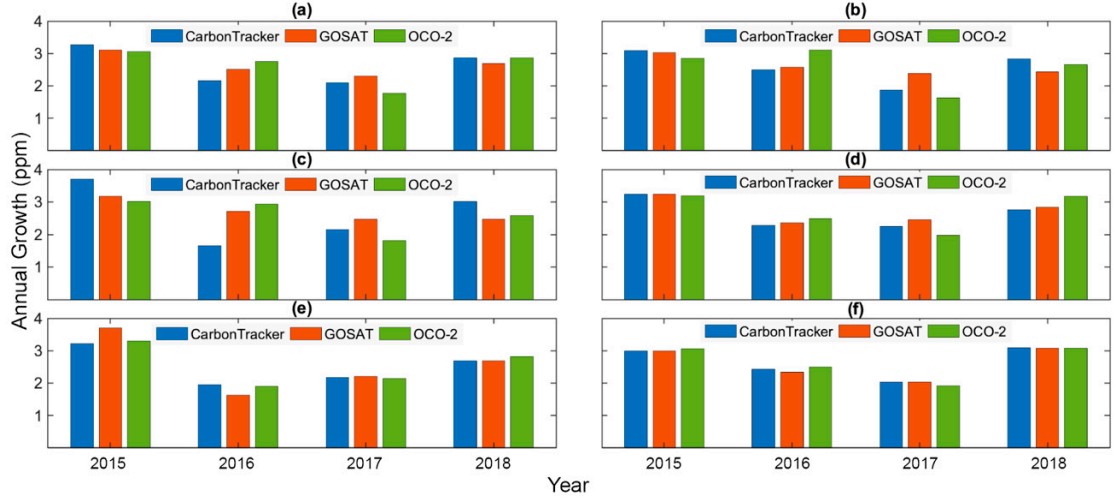

**Figure 7.** The annual growth rate of $XCO_2$ concentration for the CT and the satellite datasets over (**a**) Asia; (**b**) Central Asia; (**c**) East Asia; (**d**) South Asia; (**e**) Southeast Asia; and (**f**) Western Asia

The annual growth rate has been computed by subtracting the mean from the mean of the next year. The annual growth of $CO_2$ rate ranges from 0.98 to 3.28 ppm. It is important to note that in

all the data products, the annual growth rate is relatively higher in 2015. This is because of 2015 El Niño which occurred in March 2015. El Niño–Southern Oscillation (ENSO), characterized by anomalous sea surface warming and cooling in the eastern and central Pacific is closely related to the growth rate of atmospheric $CO_2$ [71]. Atmospheric $CO_2$ growth rate is increased during El Niño and decreases during La Nina [72]. The computed growth rates are in agreement with the NOAA growth rates from $CO_2$ surface observations [5] and the World Data Center for Greenhouse Gases (WDCGG) (https://gaw.kishou.go.jp/publications/global_mean_mole_fractions).

### 3.1.2. Seasonal Climatology Comparison

Figure 8 shows the seasonal variations of $XCO_2$ derived from the CT and GOSAT over Asia for a period of 10 years staring from September 2009 to August 2019 for four seasons based on the three months; winter (December January February or DJF), spring (March April May or MAM), summer (June July August or JJA) and autumn (September October November or SON). The seasonal cycle has significant effects on $CO_2$ concentration.

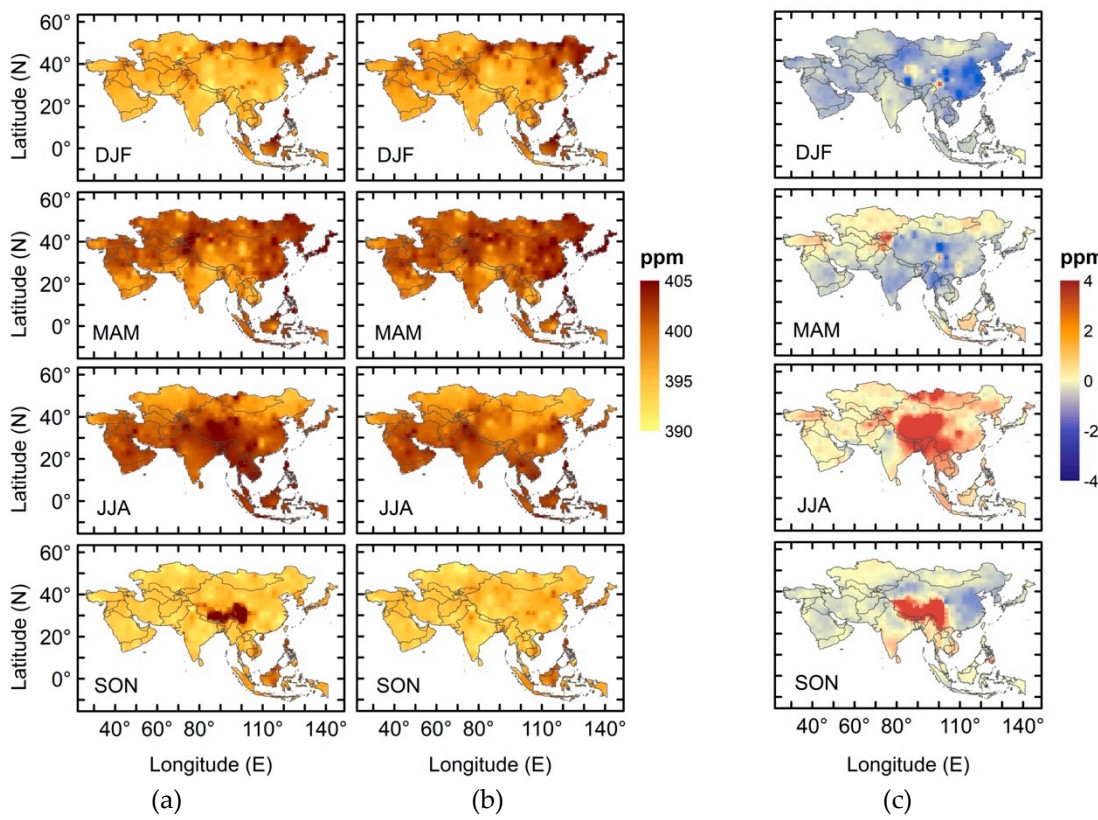

**Figure 8.** Seasonal distribution of $XCO_2$ from CT (**a**), GOSAT (**b**), and their differences (CT-GOSAT) (**c**).

The higher $CO_2$ concentrations have been observed in winter and spring while the lower ones occurred in summer and autumn. This trend is common in the Northern Hemisphere [33]. The weak photosynthesis, microorganism activity, and the heating measures are the major causes of the increased $CO_2$ concentrations in winter and the spring; while $CO_2$ uptake from vegetation in summer removes $CO_2$ from the atmosphere and decreases the $CO_2$ concentrations. In winter, spring, and autumn, the CT $XCO_2$ is smaller than that of GOSAT (Figure 8). During winter, the difference between the two datasets is significant and it reaches over −2 ppm in western China and the western part of Central Asia. Kog et al. [56] compared the seasonal climatology of CT2017 with GOSAT and found that the differences in most of the regions were within ±2.5 ppm. In the summer, the CT shows higher $CO_2$ concentrations relative to GOSAT. It is important to note that during this season, the monsoon is active

in Asian regions. Nalini et al. [70] also found that the CT shows higher concentrations over Indian regions during the monsoon. More detail about seasonal climatology comparison between the two datasets over Asian regions is given in the Table 8.

**Table 8.** The table shows the seasonal statistics of the CT and GOSAT for 10 years over Asia and its regions. The statistics are based on the seasonal average of the difference between the CT and GOSAT (D), correlation, Route Mean Square Deviation (RMSD), and the standard deviation.

| Region | Season | D (ppm) | R | RMSD | CT2019 Std | GOSAT Std | Dataset Quantity |
|---|---|---|---|---|---|---|---|
| Asia | DJF | −1.59 | 0.93 | 2.90 | 3.03 | 3.15 | 113,260 |
|  | MAM | −0.61 | 0.94 | 2.49 | 2.85 | 2.60 | 78,882 |
|  | JJA | 0.91 | 0.92 | 2.95 | 3.25 | 3.12 | 78,543 |
|  | SON | −0.49 | 0.94 | 2.65 | 2.42 | 2.56 | 150,676 |
| Central Asia | DJF | −1.21 | 0.92 | 2.23 | 2.92 | 3.01 | 8150 |
|  | MAM | 0.33 | 0.95 | 0.95 | 2.56 | 2.22 | 22,990 |
|  | JJA | 0.31 | 0.94 | 2.05 | 2.60 | 2.55 | 34,231 |
|  | SON | −0.18 | 0.98 | 1.48 | 1.99 | 2.23 | 38,009 |
| East Asia | DJF | −2.01 | 0.89 | 3.24 | 3.48 | 3.48 | 32,194 |
|  | MAM | −0.90 | 0.90 | 3.12 | 2.67 | 2.48 | 24,525 |
|  | JJA | 1.19 | 0.88 | 3.39 | 3.53 | 3.58 | 19,237 |
|  | SON | −0.88 | 0.90 | 3.06 | 2.43 | 2.56 | 55,023 |
| South Asia | DJF | −1.01 | 0.95 | 2.18 | 2.72 | 2.85 | 30,398 |
|  | MAM | −1.07 | 0.96 | 2.38 | 3.02 | 2.58 | 14,247 |
|  | JJA | 0.45 | 0.93 | 2.88 | 2.89 | 2.70 | 5614 |
|  | SON | −0.19 | 0.94 | 2.08 | 2.46 | 2.60 | 20,781 |
| Southeast Asia | DJF | −1.22 | 0.95 | 2.30 | 2.79 | 2.77 | 10,066 |
|  | MAM | −0.52 | 0.95 | 2.10 | 3.50 | 3.19 | 3853 |
|  | JJA | 1.00 | 0.96 | 1.83 | 2.39 | 2.42 | 1164 |
|  | SON | 0.21 | 0.95 | 1.94 | 2.68 | 2.87 | 2531 |
| Western Asia | DJF | −1.58 | 0.96 | 2.39 | 1.49 | 1.48 | 32,452 |
|  | MAM | −0.45 | 0.97 | 1.95 | 2.10 | 1.95 | 13,267 |
|  | JJA | 0.24 | 0.96 | 1.86 | 2.23 | 2.41 | 18,297 |
|  | SON | −0.55 | 0.97 | 1.65 | 1.58 | 1.59 | 34,332 |

　　Figure 9 shows the spatial distribution of seasonal averaged $XCO_2$ from the CT (left panel), OCO-2 (middle panel), and their differences (right panel) over Asia for a period of 5 years ranging from September 2014 to August 2019. Spatial distribution characteristics are consistent between CT and OCO-2. $XCO_2$ increases from winter (DJF) to spring (MAM), with the highest concentration observed during spring and decreases from summer (JJA) to autumn (SON), with the lowest concentration during SON. The decreasing trend from MAM to JJA to SON has likely resulted from the vegetation awakening as the photosynthesis from the vegetation leads to decaying of the $CO_2$ concentration. The decay of vegetation in autumn and the heating measures in winter result in increasing the $CO_2$ concentration. Figure 9 (right panel) displays the seasonal mean differences of $XCO_2$ from the two datasets. Both of the datasets show good agreement with each other. In most of the regions of the continent, the mean seasonal difference ranges from −1 to +1 ppm. East Asia is the region with the highest uncertainties in the continent. The highest mean difference (>2 ppm) and the lowest mean difference (<−2 ppm) have been observed in summer (JJA) and autumn (SON), respectively, over East Asia. This indicates that CT produces higher $XCO_2$ concentration when the vegetation cover decreases and produces the opposite when the vegetation cover is strong relative to OCO-2. The CT is overestimated during all the seasons over Asia when compared with OCO-2. More detail about seasonal climatology comparison between CT and OCO-2 datasets is given in the Table 9.

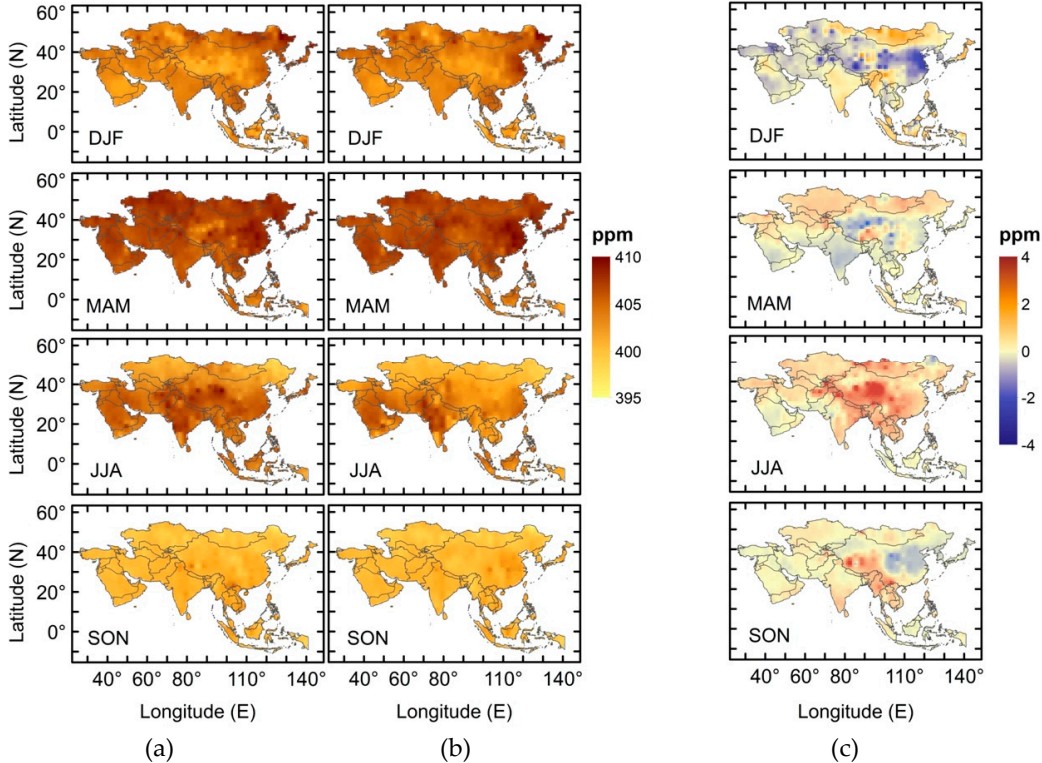

**Figure 9.** Seasonal distribution of XCO$_2$ from the CT (left panel), OCO-2 (middle panel), and their differences (CT-OCO2) (right panel).

**Table 9.** The table shows the seasonal statistics of the CT and OCO-2 for 4 years over Asia and its regions. The statistics are based on the seasonal average of the difference between the CT and OCO-2 (D), correlation, RMSD, and the standard deviation.

| Region | Season | D (ppm) | R | RMSD | CT Std | OCO-2 Std | Dataset Quantity |
|---|---|---|---|---|---|---|---|
| Asia | DJF | 0.10 | 0.85 | 2.22 | 1.69 | 1.66 | 5,895,404 |
| | MAM | 0.11 | 0.87 | 2.38 | 1.81 | 1.31 | 5,236,005 |
| | JJA | 0.82 | 0.87 | 2.40 | 2.31 | 2.23 | 6,734,436 |
| | SON | 0.10 | 0.88 | 1.95 | 1.31 | 1.25 | 7,975,485 |
| Central Asia | DJF | −0.11 | 0.89 | 2.33 | 1.73 | 2.05 | 363,017 |
| | MAM | 0.96 | 0.93 | 1.76 | 1.18 | 0.96 | 1,264,566 |
| | JJA | 1.04 | 0.90 | 2.44 | 1.99 | 1.07 | 2,442,988 |
| | SON | 0.19 | 0.95 | 1.33 | 0.88 | 0.85 | 1,607,300 |
| East Asia | DJF | −0.23 | 0.79 | 2.80 | 2.06 | 1.84 | 980,407 |
| | MAM | −0.07 | 0.80 | 2.70 | 1.63 | 1.27 | 1,465,443 |
| | JJA | 1.12 | 0.81 | 3.17 | 2.61 | 2.09 | 1,884,846 |
| | SON | −0.12 | 0.83 | 2.17 | 1.60 | 1.55 | 2,538,808 |
| South Asia | DJF | 0.02 | 0.90 | 1.80 | 0.81 | 0.58 | 2,135,507 |
| | MAM | −0.23 | 0.88 | 2.53 | 1.82 | 0.94 | 1,022,046 |
| | JJA | 1.08 | 0.84 | 2.88 | 1.91 | 2.23 | 538,615 |
| | SON | 0.37 | 0.88 | 1.88 | 0.84 | 0.97 | 1,475,048 |
| Southeast Asia | DJF | 0.45 | 0.91 | 1.64 | 1.60 | 1.75 | 539,871 |
| | MAM | 0.17 | 0.93 | 1.45 | 1.39 | 1.53 | 313,243 |
| | JJA | 0.58 | 0.93 | 1.59 | 1.46 | 1.43 | 213,821 |
| | SON | 0.43 | 0.90 | 1.71 | 1.49 | 1.43 | 240,043 |
| Western Asia | DJF | −0.32 | 0.92 | 1.44 | 0.77 | 0.90 | 1,876,602 |
| | MAM | 0.06 | 0.92 | 1.45 | 0.84 | 0.76 | 1,170,707 |
| | JJA | 0.24 | 0.93 | 1.54 | 1.26 | 1.47 | 1,654,166 |
| | SON | 0.01 | 0.93 | 1.31 | 0.55 | 0.49 | 2,114,286 |

## 4. Conclusions

This study compared the NOAA CT $XCO_2$ with that of GOSAT and OCO-2. Comparison between the CT and GOSAT has been achieved using 10 years of data ranging from September 2009 to August 2019 and the comparison between the CT and OCO-2 has been performed using 5 years of data starting from September 2014 to August 2019.

The results found that the CT $XCO_2$ is underestimated by 0.61 ppm when compared with GOSAT and overestimated by 0.31 ppm when compared with OCO-2. The differences between CT and OCO-2 are within ±1.0 ppm over most of the regions, while CT and GOSAT differ greatly as the differences between the CT and GOSAT are within ±2.0 ppm over most of the regions. Comparison between CT and satellite datasets (GOSAT and OCO-2) showed larger differences over China which may be attributed to the greater uncertainties in model terrestrial flux. The CT shows higher $CO_2$ concentrations compared to GOSAT and OCO-2 over the Tibet Plateau and the other mountain ranges.

The monthly time-series showed an agreement between CT and both GOSAT and OCO-2 in terms of amplitude and pattern. However, this agreement deteriorates over South Asia and East Asia during monsoon. The CT also showed higher $XCO_2$ concentrations and more fluctuations during the monsoon when compared with the $XCO_2$ retrieved from GOSAT and OCO-2. In the seasonal climatology comparison, the CT $XCO_2$ is more than that of OCO-2 during all the seasons whereas, the CT shows lower $XCO_2$ concentrations during autumn, winter, and spring when compared with GOSAT. The seasonal climatology between and model and the satellite datasets shows that the CT has the ability to capture the seasonal cycle.

The model and the satellite datasets have a good agreement in terms of the spatial distribution, monthly averaged distribution and seasonal climatology. These results suggest that $CO_2$ can be used from either of the datasets to understand its role in the carbon budget at a regional scale. The large uncertainties in East Asia, in particular, China are challenging. The land-use activities such as deforestation and intense agriculture also affect the $CO_2$ concentrations. Moreover, recent studies have suggested that current aerosol loading over China may affect the terrestrial carbon fluxes as well as the atmospheric $CO_2$ concentrations by diffuse radiation fertilization effects and hydrometeorological feedbacks. These effects also need to be considered in future studies.

**Author Contributions:** Formal analysis, Q.W.; Methodology, F.M.; Supervision, L.B.; Writing–review & editing, M.A.A., M.B., M.S. and Z.Q. All authors have read and agreed to the published version of the manuscript.

**Funding:** This research was funded by National Natural Science Foundation of China (NSFC), grant number 41675133 and the Special Project of Jiangsu Distinguished Professor, grant number (1421061901001) by Jiangsu Provincial Department of Education.

**Acknowledgments:** The authors acknowledge the efforts of NASA to provide the OCO-2 data products. These data were produced by the OCO-2 project at the Jet Propulsion Laboratory, California Institute of Technology, and obtained from the OCO-2 data archive maintained at the NASA Goddard Earth Science Data and Information Services Center. The authors also acknowledge NOAA Earth System Research Laboratory for their data products. The foremost author (Farhan Mustafa) is thankful to Anteneh Getachew for his support. The foremost author (Farhan Mustafa) and the 4th author (Md. Arfan Ali) are highly grateful to the China Scholarship Council (CSC) and the NUIST for granting the fellowship and providing the required supports.

**Conflicts of Interest:** The authors declare no conflict of interest.

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
