# Peer review of "Multi-Year Comparison of CO2 Concentration from NOAA Carbon Tracker Reanalysis Model with Data from GOSAT and OCO-2 over Asia"

_remotesensing, doi:10.3390/rs12152498_

Round 1
Reviewer 1 Report
The authors present a study that focuses solely on the comparison between the Carbon Tracker model and remotely sensed XCO2 from GOSAT and OCO-2, over the Asia region.
To summarise, the authors have done an adequate job in doing the comparison correctly and including the averaging kernel. The main bulk of the manuscript is a rinse-repeat formulation of some basic statistics of the satellite-model differences. At the end, the manuscript is not particularly valuable to either model scientists or XCO2 algorithm scientists - the difficulty of validation over Asia is well known, and the authors have merely pointed this fact out with figures and tables. There is nothing wrong with the comparison itself, however could be performed in a way that is of more use to the community. I have mentioned one idea further down below.
The authors do a good job presenting the data, even though the number of tables and plots might be on the large side, and I'm not sure if all figures are necessary.
My main criticisms and feedback about the manuscript are the following:
1) The colorscheme used for the various maps (blue / green / orange / red) is highly inadequate. XCO2 maps should be drawn with a sequential colormap, and difference maps that center around some value should be drawn with a diverging colormap, both should ideally be colorblind-safe. The colormaps used by authors artificially creates contrast where there is none and provides a false idea of spatial gradients and their magnitudes. I expect it is a some effort to re-create the visualizations, however I find it is important to keep in mind for future publications at least.
2) The "English" of the manuscript is good enough. There are certain sentences with faulty grammar and unusual spelling and/or wording, however it does not create any major issue for understanding the authors' intent.
3) The authors seem to consider the remotely-sensed XCO2 data as a truth, compared to CT. They do not mention the quality level of the XCO2 data itself. We know that they dependend highly on bias correction, which is in the few ppm range. It is known that especially in mountainous regions like the Tibetan pleateau, the XCO2 can be biased significantly.
Especially the last sentence in the manuscript exemplifies my issue:
"However, the discrepancies between the CT and the satellite datasets suggest that there is a need to improve the CT model data over some regions, in particular, East Asia through the assimilation of accurate observations."
The authors have not made a point as to why the XCO2 data is closer to the truth than the CT model. Is there a reason to assume that GOSAT/OCO-2 are closer to the truth than CT? Given the fact that the authors have used only one L2 product of each satellite, which use the same core algorithm - it is reasonable to assume that some fundamental biases are apparent in both GOSAT and OCO-2 XCO2.
XCO2 validation data in Asia is somewhat difficult to obtain for different reasons, and the authors mention "accurate observations" that CT should assimilate. Which particular observations do the authors have in mind?
I would recommend for the authors to work through their manuscript again and look at the comparison exercise from an additional perspective: neither CT nor GOSAT/OCO-2 are "absolute" truth data. Can they provide additional data or analysis that would show that one of the two is closer to the truth?
4) The general outline of the paper is not particularly motivating, and does not spell out a clear message. What have the authors learned from this that is important to pass onto science users of either Carbon Tracker or the OCO-2/GOSAT XCO2 products?
5) Particular suggestions for corrections:
Line 251: "The spatial distance .." (unclear meaning)
Figure 6(c): There seems to be a large outlier for the CT time series towards the beginning. Is this a code bug?
Reviewer 2 Report
The paper aims to better understanding of CO2 variation trends over Asia by comparison of XCO2 from the NOAA CarbonTracker (CT) model and two satellite (GOSAT and OCO-2) retrievals from. Additionally, monthly averaged time-series and seasonal climatology comparisons were also performed separately over the five regions of Asia; i.e., Central Asia, East Asia, South Asia, Southeast Asia, and Western Asia.
The article is written superficially, without essential details of interest to the scientific community. For example, authors are indicated that the study area is very diverse climatically, but the analysis of XCO2 does not emphasize this in any way. The reader is practically not shown the difference in XCO2 for the studied five regions of Asia. The heterogeneity of the surface or the influence of the powerful monsoons is not discussed at all.
Strange periods of study are selected. Instead of a comprehensive analysis of the three sets for the period 2014-2019, the authors are actually trying to make two independent analyzes for CT vs GOSAT, and CT vs OCO-2. Why not compare GOSAT and OCO-2 directly?
In addition, the text contains many technical inaccuracies (some of them are shown below) and is difficult to understand. Potentially, the authors have a good idea for research, but writing a paper requires more thoughtful analysis, so I suggest a major revision of the manuscript.
Specific comments:
The version of CarbonTracker is unclear: L.164 - CT2019, L.219 - CT2017.
L.19-20: Are OCO-2 retrievals available for the period from September 2009 to August 2019?
L.74: XCO2 needs to be defined in the text before the first use.
L.81-83: How to get precise in situ measurements for the satellite XCO2 validation? This sentence needs to be revised.
L.112: What does it mean “climatic continent”, “It experiences every known climate.”?
L. 126: Double definition for National Institute of Environmental Sciences (NIES)
L.137-138: The sentence is unclear. Please revise
L.159: “and” is not necessary here.
L.183-185: This explanation looks weird, as equation 1 is for calculation of Total Column CO2. Please revise
Fig.2 and further: What is the resolution of the plotted datasets?
L.244: What kind of “the selection criteria”?
L.245: What does it mean “relative spatial distance”?
Fig. 4. Are you sure the fitting line is correct for the (b) panel?
L.251-252: What does it mean? What kind of discrepancy?
L.313-314: The sentence is vague.
L.374-386: Two figures #10.
L.385: all panels of the figure labeled as (a)
L.387: For which time period the seasonal climatology is derived?
Reviewer 3 Report
Dear Editor for the Journal of Remote Sensing
While it is clear that a significant amount of work went into this effort, it is somewhat unclear as to the scientific objectives of this article. The comparison of multi year reanalysis data to remote sensed data from one or more on orbit instruments is always a very delicate matter, considering it is hard to identify the "truth"/"gold-standard" to determine which provides the best unbiased view of the short/long-term atmospheric state. That being said, I think the article at a minimum needs
1) Introduction needs to be strengthen, and the authors may consider exploring the similarities and difference between the on-orbit calibration mechanisms, inputs to the retrieval and those use as significant drivers for CT. Figure 4 seems to visually indicate a more coupled relationship between GOSAT and CT, and the between OCO-2 and CT. Is this relevant?
2) The authors should consider explicitly stating the significant of each metric used in the study and their relevance to the analyses. In addition a sentence or two about the AP errors and their significance should be explained.
3) The graph should be consolidated into summary figures or examples for those metric or information that support the analysis. There seems to be a number of figures that consume a considerable amount of page space with very little support of the study results.
4) The author might consider the use of words like bias and precision, that often have reference to a know true value.
5) It might be interesting to compute a bounds based on the differences between GOSAT and OCO-2 retrievals, in an effort not compare the two, but to construct an error envelope to understand if the observed differences are statistically significant.
5) The conclusion should be reconsidered, or maybe rephrased with some supporting evidence as to why the remote sensing data is considered to be more representative of the "truth" than CT. and what might drive on or the other to be biased in some way shape or form. Is it possible that CT is more representative of the "truth" and the remote sensing retrieval process should be revisited? What might drive these difference in Asia from both a retrieval and reanalysis perspective?
Addition comments have been embedded in the attached document along with some suggest rephrasing.
Round 2
Reviewer 2 Report
Although the article has been significantly improved as a result of the revision, I must note that not all comments have been addressed. My greatest concern is still not revised: "the authors are actually trying to make two independent analyzes for CT vs GOSAT, and CT vs OCO-2. Why not compare GOSAT and OCO-2 directly? " I believe that Figures 6-7 and 8-9 should be combined to improve readability and reduce the size of the article.
After the major revision with multiple improvements, the draft became difficult to read. Please, provide the final version of the article along with a version tracking the corrections.
Author Response
Dear Sir,
Please see the attachment. I have attached response along with the final version of the manuscript. I cannot upload the track changes version of the manuscript here because only 1 file is allowed to be uploaded. I will request the editor to send you the track changes version of the manuscript.
Kind regards

Reviewer 3 Report
A significant amount of work seems to have gone into constructing a more balanced manuscript. Overall I think with some further refinements this manuscript would be acceptable for publication. I might suggest an additional pass a the english grammar and a check for things like subscripts on CO2 and XCO2, and potentially changing "the GOSAT" and "the OCO-2" to simply GOSAT and OCO-2 in particular. Finally, I still feel that the authors might consider strengthen their conclusion, which I believe has going in the right direction but still may need some more observations, concluding ideas, where it may and may not be best suite over Asia, which regions are of concern and what might be done to refine our understanding in the future.
